# Energetic Features of H-Bonded and π-Stacked Assemblies in Pyrazole-Based Coordination Compounds of Mn(II) and Cu(II): Experimental and Theoretical Studies

**Mridul Boro** [1], **Trishnajyoti Baishya** [1], **Antonio Frontera** [2,*], **Miquel Barceló-Oliver** [2] and **Manjit K. Bhattacharyya** [1,*]

[1] Department of Chemistry, Cotton University, Guwahati 781001, Assam, India; boromridul8@gmail.com (M.B.); baishyatrishnajyoti@gmail.com (T.B.)

[2] Departament de Química, Universitat de les Illes Balears, Crta de Valldemossa km 7.7, 07122 Palma de Mallorca, Baleares, Spain; miquel.barcelo@uib.es

[*] Correspondence: toni.frontera@uib.es (A.F.); manjit.bhattacharyya@cottonuniversity.ac.in (M.K.B.)

**Abstract:** Two new coordination compounds comprising Mn(II) and Cu(II) viz. [Mn(bz)$_2$(Hdmpz)$_2$(H$_2$O)] (**1**) and [Cu(crot)$_2$(Hdmpz)$_2$] (**2**) (where, bz = benzoate; crot = crotonate; Hdmpz = 3, 5-dimethyl pyrazole) were synthesized and characterized. The characterization involved a single crystal X-ray diffraction technique, FT-IR spectroscopy, electronic spectroscopy, TGA, and elemental analyses. Compounds **1** and **2** crystallize as mononuclear entities of *Hdmpz* with penta-coordinated Mn(II) and hexa-coordinated Cu(II), respectively. These complexes exhibit distorted trigonal bipyramidal and distorted octahedral geometries, respectively. A crystal structure analysis of compound **1** elucidates the existence of C–H···π and π-stacking interactions alongside O–H···O, N–H···O, and C–H···O H-bonding interactions contributing to the stabilization of the compound's layered assembly. Similarly, in compound **2**, the crystal structure stability is attributed to the presence of hydrogen bonding in conjugation with π-stacking interactions. We conducted theoretical investigations to analyze π···π, H-bonding, and antiparallel CH···π non-covalent interactions observed in compounds **1** and **2**. DFT calculations were performed to find out the strength of these interactions energetically. Moreover, QTAIM and non-covalent interaction (NCI) plot index theoretical tools were employed to characterize them and evaluate the contribution of the H-bonds.

**Keywords:** mononuclear coordination compound; aromatic π-stacking; DFT; QTAIM; NCI

## 1. Introduction

In recent years, there has been notable vigor in the advancement of mixed ligand metal–organic frameworks and supramolecular architectures, owing to their expansive potential utility across magnetic devices, non-linear optics, biology, sorption, sensors, electrical conductivity, and catalysis [1–7]. Yet, the synthesis and fabrication of single crystals with desired architectures and envisioned functionalities continue to pose significant challenges. This is largely attributed to the intricate self-assembly processes, which are highly contingent upon various experimental variables including the coordination environment of the central metal ion, the specific nature of the ligands utilized, the reaction condition, the metal-to-ligand ratio, and environmental factors [8–12]. Effective integration of these synthetic parameters is paramount in realizing a desired supramolecular architecture with significant potential [13].

The discernment of non-covalent interactions is fundamental in crafting self-assembled architectures. These interactions lay the groundwork for achieving precise recognition, transport, and regulatory capabilities with remarkable specificity [14–16]. The profound alternations in the properties of self-assembled molecules arise from the nuanced interplay of non-covalent interactions, including aromatic π-stacking, C–H···π interactions, and a

diverse array of hydrogen bonding interactions. These non-covalent interactions dictate the directionality and collective potency, serving as a driving force for the organization of molecules into intriguing self-assembled structures [17–22]. In addition, non-covalent interactions are of profound importance in molecular biology and involve drug–receptor interactions, protein folding, etc. [23–25]. To date, there have been considerable efforts to explore and quantify the array of non-covalent interactions observed in self-assembled architectures [26,27].

The focus of chemists has now shifted towards N and O donor ligands, producingsubstantial interest in molecular biology. These ligands serve as key constituents in crafting self-assembled architectures, offering multifaceted utility across a broad array of applications [28,29]. Transition metal coordination complexes involving aliphatic and aromatic carboxylates offer diverse and captivating structural networks, as they possess the capability to connect through monodentate symmetric and asymmetric chelating and bidentate and monodentate bridging coordination modes, adding to their intricate nature [30,31].The aromatic benzoate group, when utilized as a substituent, acts as a complex targeting moiety facilitating the delivery of the metal compound to bacterial cells within the body [32]. Similarly, metal crotonate complexes also display different biological activities like DNA binding, radical scavenging activities, etc. [33]. In a similar way, pyrazole ligand complexes, with their diverse structural configurations, exhibit significant efficacy and versatility across various fields, including medicine, catalysis, separations, bio-mimetic chemistry, optics, magnetism, and luminescence [34–37]. Pyrazole complexes, exhibiting diverse pharmacological activities including antifungal [38,39], antibacterial [40], and anticancer properties, have been identified as pivotal contributors in the development of innovative drugs [41–44]. Furthermore, there is growing interest in the coordination chemistry of transition metal-based drugs, driven by their potential applications in cancer treatment [45–47]. Transition metals like manganese are integral to numerous vital biological processes, encompassing electron transfer, catalysis, and structural functions. They are prevalent in the active sites of numerous proteins and enzymes, playing pivotal roles in facilitating their activities [48]. Moreover, Mn(II) complexes demonstrate compelling electrochemical, biological, and magnetic properties [49,50]. The innate transition elemental characteristics of the Cu(II) ion amplify its coordination propensities towards aromatic and aliphatic ligands, thus accentuating its notable flexibility [51]. N- and O-donor ligand copper complexes with crystal structures are reported in the literature [52,53]. On the basis of ligand binding sites, some Cu(II) complexes possess numerous biological activities, such as antibacterial, fungicidal, pesticidal, and even tracer activities [54,55].

With an aim to explore the cooperation of non-covalent interactions in supramolecular architectures, two novel Mn(II) and Cu(II) coordination compounds viz. [Mn(bz)$_2$(Hdmpz)$_2$(H$_2$O)] (**1**) and [Cu(crot)$_2$(Hdmpz)$_2$] (**2**) were synthesized and meticulously characterized utilizing a single crystal X-ray diffraction technique, FT-IR, electronic spectroscopy, TGA, and elemental analysis to explore the interplay of non-covalent interactions within supramolecular architectures. The crystal structure analysis of compound 1 revealed the presence of C–H···π and π-stacking interactions along with O–H···O, N–H···O, and C–H···O H-bonding, contributing to the stability of its layered assembly. In compound 2, aromatic π-stacking interactions, C–H···O and N–H···O hydrogen bonds along with non-covalent C–H···C interactions were identified as key stabilizing forces in its crystal structure. We conducted theoretical studies to examine the π-stacking, H-bonds, and C–H···π interactions present in compounds 1 and 2. These interactions were analyzed utilizing the quantum theory of atoms-in-molecules (QTAIM) and the non-covalent interaction (NCI) plot index computational tools.

## 2. Materials and Methods

All the chemicals, viz. manganese (II) chloride tetrahydrate, copper(II) chloride dihydrate, benzoic acid, 3,5-dimethyl pyrazole, and crotonic acid, used for synthesis were obtained from commercial sources and were used as received. Elemental analyses (C, H,

N) were performed with the Perkin Elmer 2400 series II CHN analyzer. Infrared spectra (4000–500 cm$^{-1}$) were recorded with a Bruker Alpha (II) Infrared spectrophotometer on samples of compound **1** and **2** prepared as KBr pellets. A Shimadzu UV-2600 spectrophotometer was used to record the electronic spectra of the compounds. BaSO$_4$ powder was used as a reference (100% reflectance) to record the solid-state spectra. Magnetic moments of the compounds were measured at room temperature (300K) using the Evans method with Sherwood Mark 1 Magnetic Susceptibility balance. For understanding the thermal stability of our compounds, thermogravimetric analysis was conducted under a dinitrogen atmosphere using a Mettler Toledo TGA/DSC1 STAR$^e$ system at a heating rate of 10 °C min$^{-1}$.

### 2.1. Synthesis

#### 2.1.1. Synthesis of [Mn(bz)$_2$(Hdmpz)$_2$(H$_2$O)] (**1**)

An aqueous solution (5 mL) of Hdmpz (0.192 g, 2 mmol) was poured slowly after two hours into a deionized water solution (10 mL) containing MnCl$_2$·4H$_2$O (0.197 g, 1 mmol) and the sodium salt of benzoic acid (0.288 g, 2 mmol) and it was kept stirring for another hour (Scheme 1). The resulting yellow solution was left undisturbed, and yellow block-shaped single crystals were obtained after a few days by slow evaporation in aqueous medium (2–4 °C). Yield: 0.465 g (92.26%). Anal. calcd. for C$_{24}$H$_{28}$MnN$_4$O$_5$ C, 56.81%; H, 5.56%; N, 11.04%; Found: C, 55.90%; H, 5.47%; N, 10.99%. FT-IR (KBr pellet, cm$^{-1}$): 3439 (br), 3130 (w), 2837 (m), 1593 (s), 1429 (m), 1389 (s), 1280 (m), 1145 (m), 1108 (w), 975 (w), 772 (m), 715 (m), 655 (s) (s, strong; m, medium; w, weak; br, broad; sh, shoulder).

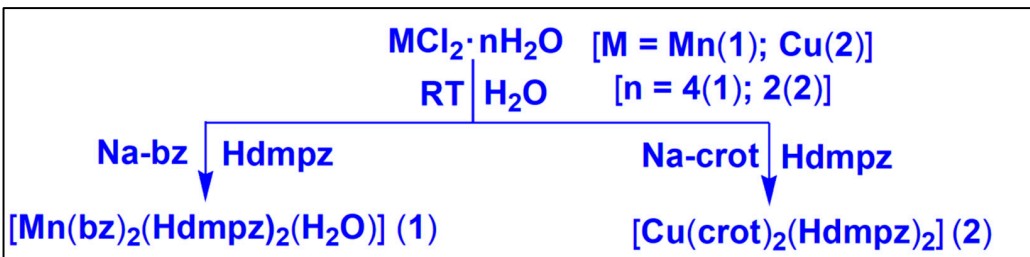

**Scheme 1.** Synthesis of the compounds **1** and **2**.

#### 2.1.2. Synthesis of [Cu(crot)$_2$(Hdmpz)$_2$] (**2**)

Compound **2** was obtained by applying similar procedures as those used for compound **1**, but with CuCl$_2$·2H$_2$O (0.170 g, 1 mmol) and the sodium salt of crotonic acid (0.210 g, 2 mmol) instead of MnCl$_2$·4H$_2$O and the sodium salt of benzoic acid (Scheme 1). The resulting blue solution was kept undisturbed, and blue block-shaped single crystals were obtained by slow solvent evaporation in a refrigerator (below 4 °C). Yield: 0.386 g (90.82%). Anal. calcd. for C$_{18}$H$_{26}$CuN$_4$O$_4$: C, 50.75%; H, 6.15%; N, 13.15%; Found: C,50.67%; H, 6.09%; N, 13.09%. IR (KBr pellet, cm$^{-1}$): 3439 (br), 3132 (sh), 2845 (m), 1593 (s), 1430 (m), 1414 (s), 1288 (m), 1150 (m), 1115 (w), 1045 (m), 944 (w), 849 (m), 740 (m), 670 (m), 498 (w) (s, strong; m, medium; w, weak; br, broad; sh, shoulder).

### 2.2. Crystallographic Data Collection and Refinement

Initial crystal evaluation and data collection were performed at 100K using graphite mono-chromatized Cu/Kα radiation, λ = 1.54178 Å with a Bruker APEX-II CCD diffractometer.Multiscan absorption correction in addition to the scaling and merging of the various datasets for the wavelength were carried out using SADABS [56]. The crystal structures were solved using a direct method and refined by full-matrix least-squares procedures, based on F$^2$ with all measured reflections, with SHELXL-2018/3 [57] using the WinGX [58] software. All non-hydrogen atoms were refined anisotropically. The hydrogen atoms except those attached to the O-atoms of water molecules were introduced their idealized positions and refined in the isotropic approximation. The hydrogen atoms of the

coordinated water molecules were fixed at the nominal X-ray distances from the O atoms to obtain the hydrogen bonding patterns in the crystal structures. Diamond 3.2 [59] software was used to generate the structural drawings for this publication. Crystallographic data of compounds **1** and **2** are tabulated in Table 1.

**Table 1.** Crystallographic data and structure refinement details for compounds **1** and **2**.

| Crystal Parameters | 1 | 2 |
|---|---|---|
| Empirical formula | $C_{24}H_{28}MnN_4O_5$ | $C_{18}H_{26}CuN_4O_4$ |
| Formula weight | 507.44 | 425.97 |
| Temperature (K) | 100.0 | 100.0 |
| Wavelength (Å) | 1.54178 | 1.54178 |
| Crystal system | Orthorhombic | Orthorhombic |
| Space group | Pbcn | Pbcn |
| $a$/Å | 19.2401(1) | 15.6899(5) |
| $b$/Å | 12.1579(7) | 10.7161(3) |
| $c$/Å | 10.1312(6) | 11.3062(4) |
| $\alpha$ ° | 90 | 90 |
| $\beta$ ° | 90 | 90 |
| $\gamma$ ° | 90 | 90 |
| Volume (Å$^3$) | 2369.9(2) | 1900.96(1) |
| Z | 4 | 4 |
| Calculated density (g/cm$^3$) | 1.422 | 1.488 |
| Absorption coefficient (mm$^{-1}$) | 4.893 | 1.897 |
| F(000) | 1060 | 892 |
| Crystal size (mm$^3$) | $0.35 \times 0.21 \times 0.18$ | $0.31 \times 0.22 \times 0.12$ |
| $\theta$ range for data collection (°) | 10.76 to 136.33 | 1.604 to 23.973 |
| Index ranges | $-23 \le h \le 23,$ $-14 \le k \le 14,$ $-11 \le l \le 12$ | $-18 \le h \le 18,$ $-12 \le k \le 12,$ $-13 \le l \le 13$ |
| Reflections collected | 19,422 | 28,974 |
| Unique data ($R_{int}$) | 2972 | 1721 |
| Refinement method | Full-matrix least-squares on F$^2$ | Full-matrix least-squares on F$^2$ |
| Data/restraints/parameters | 2136/1/161 | 1721/0/126 |
| Goodness-of-fit on $F^2$ | 1.086 | 1.143 |
| Final *R*indices[$I > 2\sigma$ (I)] $R_1/wR_2$ | $R_1 = 0.0414$, wR$_2$ = '0.1110 | $R_1 = 0.0475$, wR$_2$ = 0.1395 |
| *R*indices (all data) $R_1/wR_2$ | $R_1 = 0.0434$, wR$_2$ = 0.1133 | $R_1 = 0.0482$, wR$_2$ = 0.1402 |
| Largest diff. peak and hole (e.Å$^{-3}$) | $0.45/-0.49$ | $0.37/-0.63$ |

CCDC 2,261,049 and 2,261,050 contain the supplementary crystallographic data for compounds **1** and **2**, respectively. These data can be obtained free of charge at http://www.ccdc.cam.ac.uk or from the Cambridge Crystallographic Data Centre, 12 Union Road, Cambridge CB2 1EZ, UK; fax: (+44) 1223-336-033; or e-mail: depo-sit@ccdc.cam.ac.uk.

### 2.3. Computational Methods

Single point calculations were performed using the Gaussian-16 software suite [60], employing the PBE0 [61]-D3 [62]/def2-TZVP [63] level of theory. Crystallographic coordinates were utilized to assess the interactions within the compounds, with a specific focus on the non-covalent interactions prevalent in their solid state. To analyze these interactions, Bader's "Atoms in Molecules" (QTAIM) theory [64] and the non-covalent interaction (NCI) plot [65,66] technique were applied, using the AIMAll software [67]. The hydrogen bond energies were determined based on the formula proposed by Espinosa et al. (E = ½Vr) [68].

## 3. Results and Discussion

### 3.1. Syntheses and General Aspects

[Mn(bz)$_2$(Hdmpz)$_2$(H$_2$O)] (**1**) was synthesized by reacting one equivalent of MnCl$_2$·4H$_2$O with two equivalents of sodium salt of benzoic acid and two equivalents

of Hdmpz at ambient temperature. In parallel, [Cu(crot)$_2$(Hdmpz)$_2$] (**2**) was prepared by reacting one equivalent of CuCl$_2$·2H$_2$O, two equivalents of the sodium salt of crotonic acid and two equivalents of Hdmpz at room temperature in de-ionized water medium. Both **1** and **2** are soluble in water and in common organic media. Compounds **1** and **2** display room temperature (298 K) $\mu_{eff}$ values of 5.89 and 1.82 BM, respectively, to indicate the presence of five and one unpaired electron(s) in the Mn(II) and Cu(II) centers of the distorted trigonal bipyramidal and distorted octahedral coordination spheres [69,70].

### 3.2. Crystal Structure Analysis

Figure 1 depicts the molecular structure of **1**. Compound **1** grows in an orthorhombic crystal system with a Pbcn space group. Selected bond lengths and bond angles are summarized in Table 2.

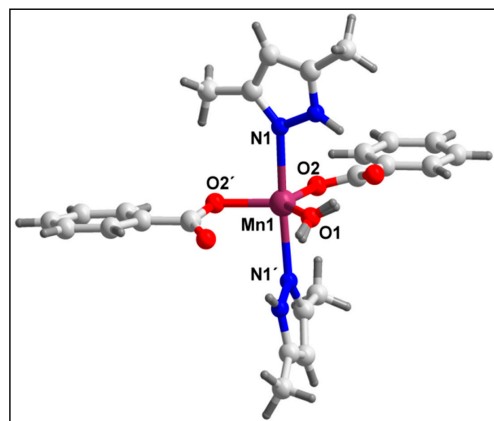

**Figure 1.** Molecular structure of [Mn(bz)$_2$(Hdmpz)$_2$(H$_2$O)] (**1**).

**Table 2.** Selected bond lengths (Å) and bond angles (°) around the Mn(II) centers in **1** and **2**, respectively.

| Bond Lengths of 1 (Å) | | Bond Angles of 1 (°) | |
|---|---|---|---|
| Mn1–O2 | 2.0854(1) | O2–Mn1–O2´ | 106.18(8) |
| Mn1–O2´ | 2.0854(1) | O2–Mn1–O1 | 126.91(4) |
| Mn1–O1 | 2.132(2) | O2´–Mn1–O1 | 126.91(4) |
| Mn1–N1 | 2.3173(2) | O2–Mn1–N1´ | 92.39(5) |
| Mn1–N1´ | 2.3173(2) | O2–Mn1–N1 | 91.72(6) |
| | | O1–Mn1–N1 | 86.57(4) |
| | | O1–Mn1–N1´ | 86.57(4) |
| | | O2´–Mn1–N1´ | 91.72(6) |
| | | O2´–Mn1–N1 | 92.40(5) |
| | | N1–Mn1–N1´ | 173.15(8) |
| **Bond Lengths of 2 (Å)** | | **Bond Angles of 2 (°)** | |
| Cu1–N1 | 1.989(3) | N1–Cu1–N1 | ´93.24(1) |
| Cu1–N1´ | 1.989(3) | N1–Cu1–O1 | 165.45(9) |
| Cu1–O1´ | 2.002(2) | N1–Cu1–O1´ | 91.47(9) |
| Cu1–O1 | 2.002(2) | N1´–Cu1–O1 | 91.47(9) |
| Cu1–O2 | 2.492(2) | N1´–Cu1–O1´ | 165.45(9) |
| Cu1–O2´ | 2.492(2) | O1–Cu1–O1´ | 87.34(1) |

In the compound, the Mn(II) metal center is penta-coordinated to two monodentate bz moieties, two monodentate Hdmpz, and one water molecule. The coordination geometry around the Mn1 center in the compound is slightly distorted trigonal bipyramidal as evident from the trigonality index value ($\tau$) of 0.77 [71], where the axial sites are occupied by N1 and N1' atoms from Hdmpz moieties while the equatorial sites are occupied by O1, O2, and O2´ from coordinated water and bz moieties, respectively. The bond lengths between the Mn1 and the nitrogen atoms of Hdmpz (N1 and N1´) are found to be 2.32(2) Å; however, that between the Mn1 and oxygen atoms of bz moieties (O2 and O2´) is 2.08(1) Å. The bond length between the Mn1 center and the oxygen atom (O1) of the coordinated water molecule was found to be 2.13Å. Crystal structure analysis reveals that hydrogen atoms (H1A and H1B) of the coordinated water molecule have a site occupancy factor of 0.5. The average Mn–O and Mn–N bond distances are almost consistent with the previously reported Mn(II) complexes [72,73].

The neighboring monomeric units of compound **1** are linked via non-covalent C–H$\cdots\pi$ and $\pi$-stacking interactions to constitute the 1D supramolecular chain along the crystallographic c direction (Figure S1). The –CH moiety (–C4H4) of the bz moiety is involved in C–H$\cdots\pi$ interactions with the $\pi$-system of the aromatic ring of Hdmpz having a centroid (C9, C10, C11, N1, and N2)$\cdots$H4 distance of 2.70 Å. The angle between H4, the centroid of the pyrazole ring, and the aromatic plane is 159.1°, which indicates the strong nature of the interaction [74].

Aromatic $\pi$-stacking interactions are found to be present between the aromatic rings of bz moieties from the closest monomeric units with a centroid(C2-C7)–centroid(C2´-C7´) separation of 3.62 Å. The corresponding slipped angle (the angle between the ring normal and the vector joining the two ring centroids) is 21.5° [75]. Further analysis reveals that neighboring 1D chains, shown in Figure S1, interconnect via C–H$\cdots\pi$ and non-covalent C–H$\cdots$C interactions to form a layered assembly along the crystallographic ac plane (Figure 2).

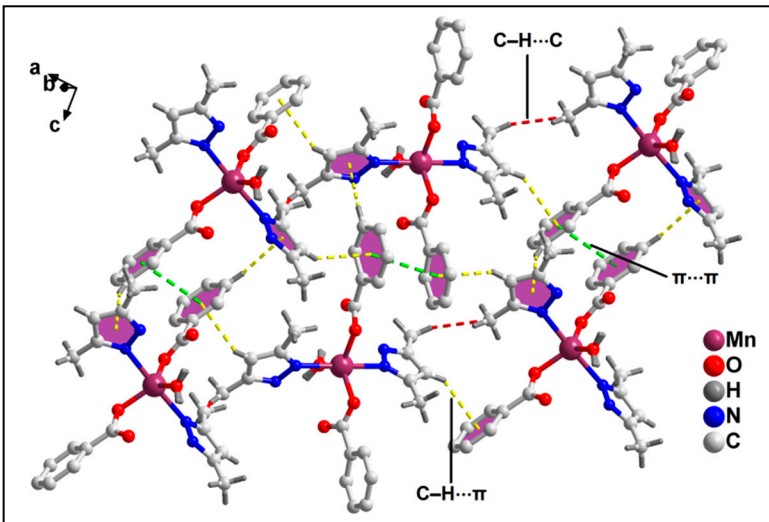

**Figure 2.** Layered assembly of compound **1** assisted by C–H$\cdots\pi$, $\pi$-stacking interactions, and non-covalent C–H$\cdots$C interactions along the crystallographic ac plane.

Extended analysis discloses the formation of one more layered architecture of the compound which is stabilized by non-covalent C–H$\cdots$C; C–H$\cdots\pi$; C–H$\cdots$O, O–H$\cdots$O, and N–H$\cdots$O hydrogen bonding and aromatic $\pi$-stacking interactions along the crystallographic bc plane (Figure 3a). Non-covalent C–H$\cdots$C interactions are present between the –CH moieties (–$C_6H_6$ and –$C_7H_7$) and C5 (from bz) and C8 (from Hdmpz) atoms from two adjacent monomeric units, having C6–H6$\cdots$C8 and C7–H7$\cdots$C5 distances of 2.94 and 3.11 Å, respectively [C6($sp^2$)–H6$\cdots$C8($sp^3$); C6$\cdots$C8 = 3.60Å; C7($sp^2$)–H7$\cdots$C5($sp^2$);

C7···C5 = 3.79Å]. The –CH moiety (–C12H12A) of Hdmpz is utilized in C–H···π interactions with the π-system of the aromatic ring of another Hdmpz moiety from a neighboring monomeric unit having a centroid (C9-C11, N1, N2)···H12A distance of 3.04 Å. The angle between H12A, the centroid of the Hdmpz moiety, and the aromatic plane is 150.2°, which evidences the high strength of the interaction [76–78]. C–H···O hydrogen bonding interactions are observed between the –C12H12A moiety of Hdmpz and the uncoordinated carboxyl atom (O3) of bz from two neighboring units having a C12–H12A···O3 distance of 2.95 Å.

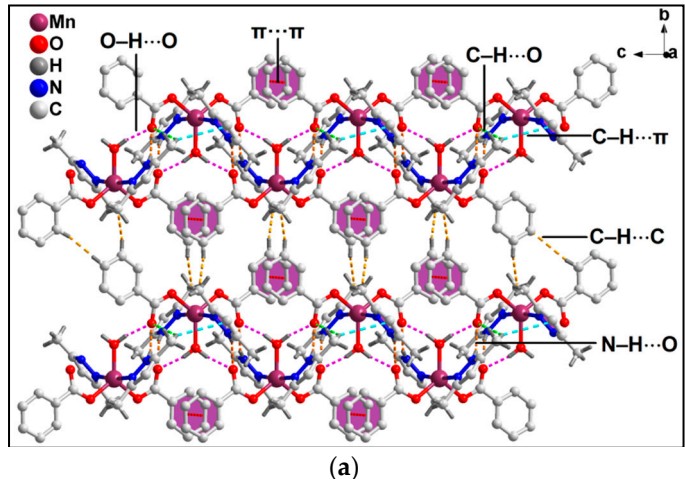

(**a**)

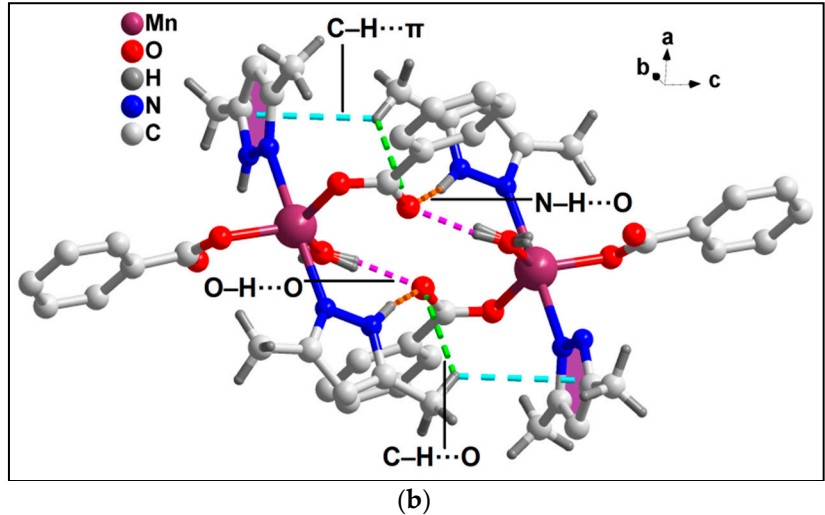

(**b**)

**Figure 3.** (**a**) Layered architecture of compound **1** having non-covalent C–H···C; C–H···π; C–H···O, O–H···O, and N–H···O hydrogen bonding and aromatic π-stacking interactions along the crystallographic bc plane; (**b**) a self-assembled dimer isolated from the layered architecture which is theoretically studied.

The O3 atom of bz moieties is also involved in N–H···O hydrogen bonding interactions with –N2H2 fragments from Hdmpz moieties of adjacent monomeric units having a N2–H2···O3 distance of 2.10 Å. Moreover, the O3 atom of bz is also engaged in O–H···O hydrogen bonding interactions, where the coordinated water molecules have a O1–H1A···O3 distance of 1.86 Å. A self-assembled dimer (Figure 3b), taken from the layered architecture along this plane, has been theoretically studied (vide infra).

Figure 4 depicts the molecular structure of compound **2**. Compound **2** crystallizes in an orthorhombic crystal system with a Pbcn space group. Selected bond lengths and bond angles are summarized in Table 2. In compound **2**, the Cu1 metal center is hexa-coordinated

to two bidentate crot moieties and two monodentate Hdmpz moieties. The coordination geometry around the metal center is a slightly distorted octahedron where the axial sites are occupied by O1´ and N1´ atoms from crot and Hdmpz moieties, respectively, while the equatorial sites are occupied by O1, O2, O2´ from crot, and N1 from Hdmpz, respectively. The average Cu–O and Cu–N bond distances are in agreement with the earlier-reported Cu(II) complexes [79,80].

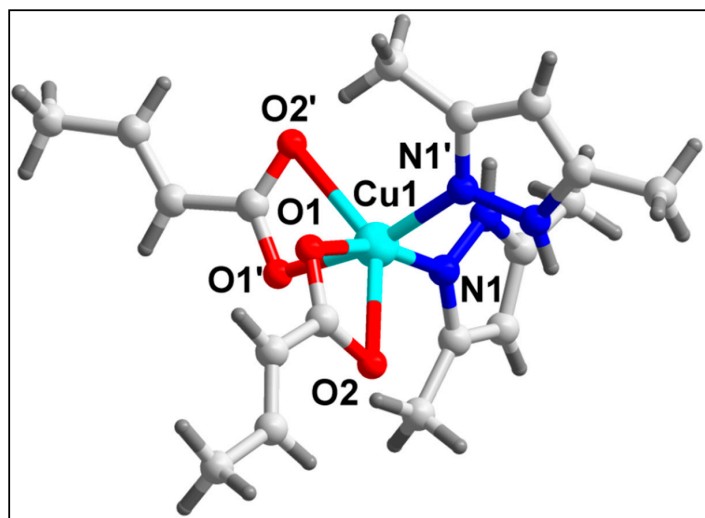

**Figure 4.** Molecular structure of [Cu(Crot)$_2$(Hdmpz)$_2$] (**2**).

The neighboring monomeric units of compound **2** connect together via non-covalent C–H$\cdots$C interactions to give rise to the 1D supramolecular chain of the compound (Figure S2). C–H$\cdots$C interactions are present between the –CH (–C9H9C and –C2H2B) and the carbon atoms (C2 and C9) of crot moieties having C9–H9C$\cdots$C2and C2–H2B$\cdots$C9 distances of 2.88 and 3.05 Å, respectively [C9(sp$^3$)–H9C$\cdots$C2(sp$^3$); C2(sp$^3$)–H2B$\cdots$C9(sp$^3$); C9$\cdots$C2 = 3.78 Å].

Moreover, two neighboring 1D chains of the compound are linked by non-covalent C–H$\cdots$C interactions, which results in the 2D layered assembly of the compound (Figure 5). C–H$\cdots$C interaction is displayed by the –CH moiety (–C3H3) of Hdmpz and the C7 atom of crot with a C3–H3$\cdots$C7 distance of 3.63 Å [C3(sp$^2$)–H3$\cdots$C7(sp$^2$); C3$\cdots$C7 = 3.86 Å].

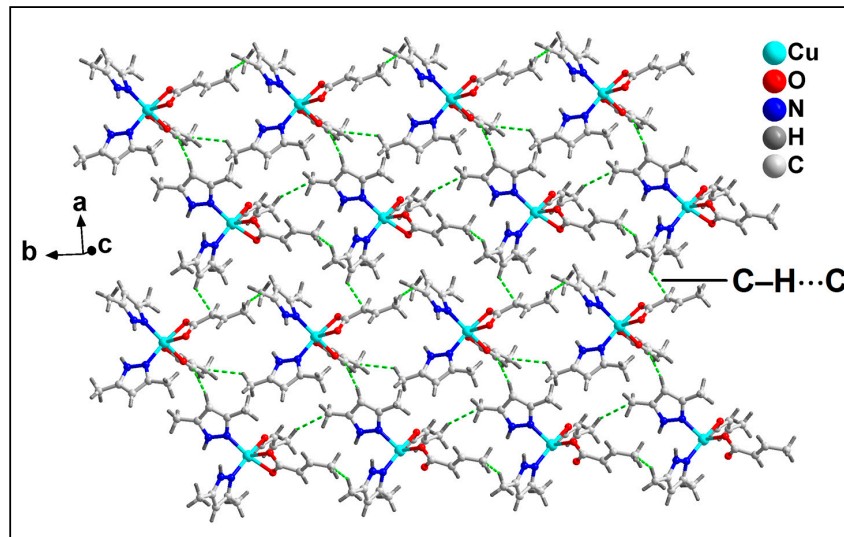

**Figure 5.** Two-dimensional assembly of compound **2** stabilized by non-covalent C–H$\cdots$C interactions along the ab plane.

Figure 6a depicts another supramolecular 1D chain of compound **2** stabilized by N–H···O hydrogen bonding and aromatic π-stacking interactions. N–H···O hydrogen bonding interaction is present between the –N2H2 moiety of Hdmpz and O2 atom of crot from two different monomeric units having a N2–H2···O2 distance of 1.90 Å. Aromatic π-stacking interactions are found between the π systems of Hdmpz and crot moieties from adjacent monomeric units of the compound with a centroid (C1, C3, C4, N1, N2)–centroid(C7-C8) distance and corresponding slipped angle of 3.92 Å and 19.1°, respectively. A self-assembled dimer (Figure 6b), retrieved from the 1D self-assembly along the crystallographic c axis, has been theoretically studied (vide infra).

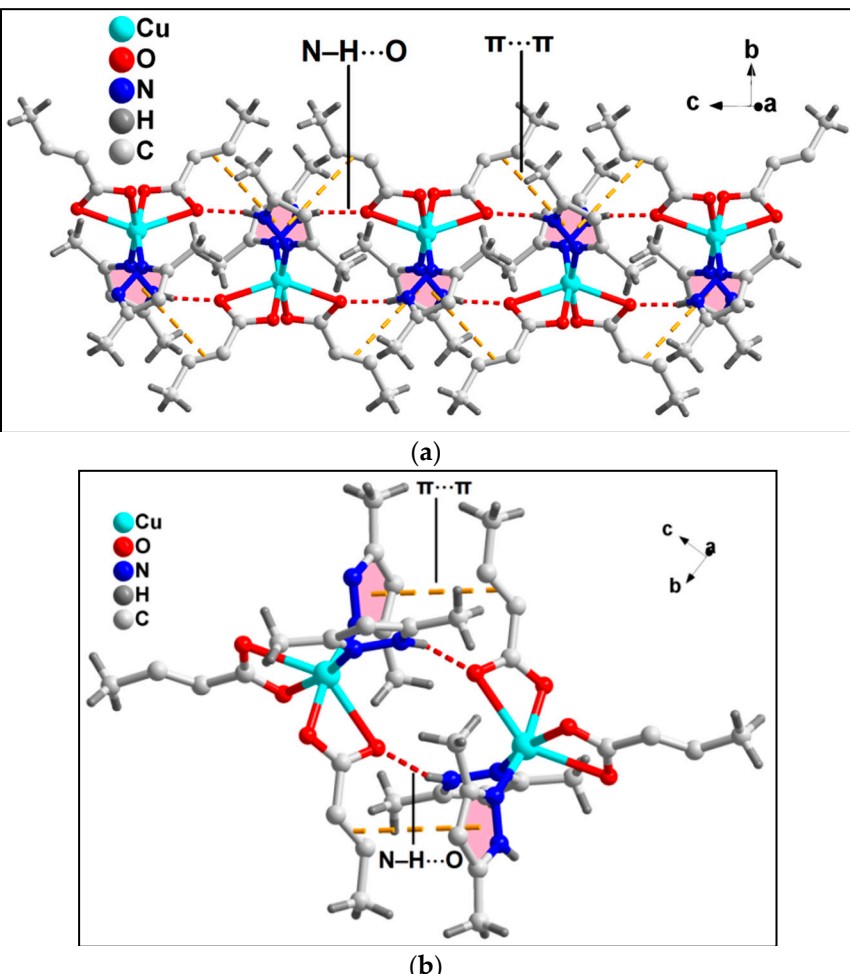

(**a**)

(**b**)

**Figure 6.** (**a**) One-dimensional supramolecular chain of compound **2** assisted by N–H···O hydrogen bonding and aromatic π-stacking interactions along the crystallographic c axis; (**b**) a self-assembled dimer obtained from the 1D chain which is theoretically studied.

These 1D supramolecular chains are interconnected via supramolecular C–H···C interactions and C–H···O hydrogen bonding to form a 2D assembly along the crystallographic bc plane (Figure 7). C–H···C interactions are present between the –C9H9C, –C7H7 fragments, and C2, C7, and C9 atoms of crot moieties from adjacent monomeric units with C9–H9C···C7,C7–H7···C9 and C9–H9C···C2 separations of 3.93, 3.97, and 2.88 Å, respectively [C9(sp$^3$); C7(sp$^2$); C2(sp$^3$); C9···C7 = 3.76 Å; C9···C2 = 3.78 Å]. Moreover, C–H···O is found to be located between the –C9H9A fragment and O1 atom of adjacent crot moieties having a C9–H9A···O1 distance of 3.07 Å (Table 3).

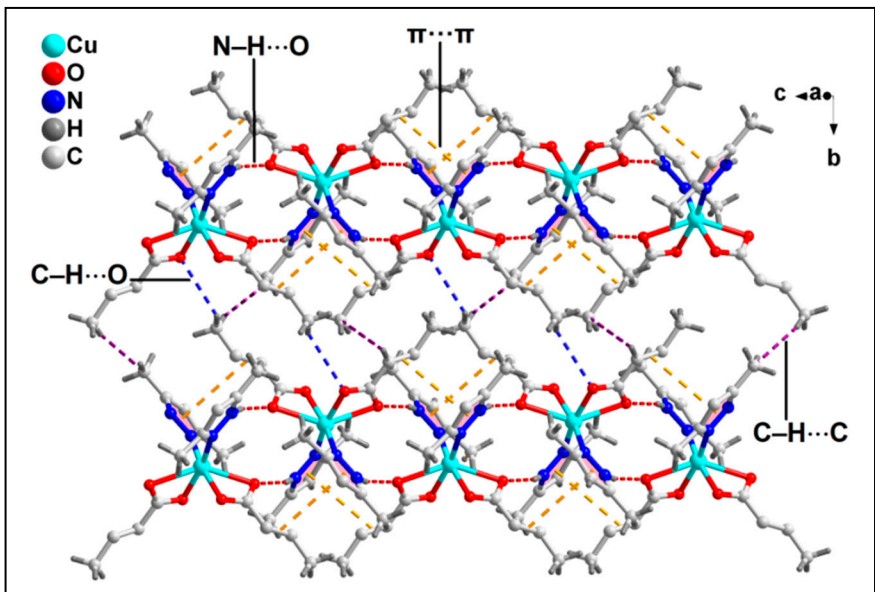

**Figure 7.** Layered architecture of compound **2** assisted by non-covalent C–H···C; N–H···O, and C–H···O hydrogen bonding and aromatic π-stacking interactions along the crystallographic bc plane.

**Table 3.** Selected hydrogen bond distances (Å) and angles (°) for **1** and **2**.

| D–H···A | d(D–H) | d(D···A) | d(H···A) | <(DHA) |
|---|---|---|---|---|
| **1** | | | | |
| C12–H12A···O3 | 0.98 | 3.64 | 2.95 | 128.6 |
| N2–H2···O3 | 0.88 | 2.91 | 2.10 | 153.7 |
| O1–H1A···O3 | 0.87 | 2.70 | 1.86 | 164.1 |
| **2** | | | | |
| N2–H2···O2 | 0.88 | 2.74 | 1.90 | 157.2 |
| C9–H9A···O1 | 0.98 | 3.57 | 3.07 | 113.2 |

*3.3. Spectral Studies*

3.3.1. FT-IR Spectroscopy

The FT-IR spectra of compounds **1** and **2** (KBr phase) were performed in the region of 4000–500 cm$^{-1}$ (Figure S3). The comparatively broad absorption peak in compound **1** at around 3440 cm$^{-1}$ can be due to the O–H stretching vibrations of the coordinated water molecule present in the compound [81,82]. Absorption bands due to $\rho_r$ (H$_2$O) (715 cm$^{-1}$) and $\rho_w$ (H$_2$O) (655 cm$^{-1}$) support the presence of coordinated water molecules in the compound [81,82]. For compound **1**, strong absorption bands appear at 1593 and 1389 cm$^{-1}$ for the asymmetric and symmetric stretching vibrations of the carboxylate groups of bz moiety, respectively [83]. The difference between the asymmetric and symmetric stretching vibrations (Δv = 204 cm$^{-1}$) of the carboxyl groups of bz moieties of **1** reflect show the carboxylate moieties are connected to the metal center in a monodentate fashion [84]. Similarly, for compound **2**, strong absorption bands appear at 1593 and 1414 cm$^{-1}$ for the asymmetric and symmetric stretching vibrations of the carboxylate groups of the crot moiety, respectively [83]. The difference between the asymmetric and symmetric stretching vibrations (Δv = 179 cm$^{-1}$) of the carboxyl groups of crot moieties of **2** indicates bidentate coordination of the carboxylate moieties to the metal center [85]. Deprotonation of the carboxyl groups upon coordination with the metal center can be identified by the absence of any sharp band at 1710 cm$^{-1}$ for both compounds **1** and **2** [86–88]. Moreover, for both the compounds, bands at around 3130 cm$^{-1}$ can be associated with the ν(N–H) vibrations of a coordinated Hdmpz ligand [89,90]. The ν(C–H) vibrations of the coordinated Hdmpz

are observed in the region of 2970–2770 cm$^{-1}$ for both the compounds [91]. The peaks at 1429, 1280, and 1145 cm$^{-1}$ in **1** can be ascribed to the C–N, N–N, and C = N stretching vibrations of Hdmpz rings, respectively; however, these peaks are obtained at 1430, 1288, and 1150 cm$^{-1}$ in the spectrum of compound **2** [92].

### 3.3.2. Electronic Spectroscopy

The electronic spectra of the compounds were performed in both solid and aqueous phases and are considered for recording the electronic spectra of **1** and **2** (Figures S4 and S5). The solid state UV-Vis-NIR spectrum of compound **1** shows no absorption bands in the visible region because of all the doubly forbidden electronic transitions from the ground state $^6A_{1g}$ of the Mn(II) center (d$^5$ system) [93]. The peaks for the π→π* absorption of the benzoate and Hdmpz ligands are obtained at the expected positions [94,95].

The solid state UV-Vis-NIR spectrum of compound **2** (Figure S5) showcases peaks at 228 and 273 nm corresponding to the π→π* transitions of the aromatic ligand [96]. The spectrum (Figure S5a) shows a broad absorption band at 741 nm resulting from the usual $^2E_g$→$^2T_{2g}$ transition for Cu(II) complexes [97]. In the UV-Vis spectra (Figure S5b) of the compound, the absorption peaks for n→π* and $^2E_g$→$^2T_{2g}$ transitions are obtained at the expected positions [97].

The similarity of absorption bands in both solid and solution phases in the spectra indicates no structural deformity of the compounds in the solution phase [98].

### 3.4. Thermogravimetric Analysis

Thermogravimetric analysis for compounds **1** and **2** was performed in between 30 and 800 °C with heating at arate of 10 °C/min under a N$_2$ atmosphere (Figure S6). In **1**, the temperature range of 120–140 °C contributes to the weight loss of coordinated water molecules (obs. = 5.1%; calcd. = 3.54%) [99,100]. In the temperature range of 141–260 °C, there is a decomposition of two benzoate moieties and one Hdmpz molecule (obs. = 65.52%; calcd. = 66.61%) [101–103]. For compound **2**, between 120–170 °C, one coordinated crot undergoes thermal decomposition with an observed weight loss of 18.40% (calcd. = 19.95%) [104]. One coordinated Hdmpz and another coordinated crot are lost in 171–290 °C having a weight loss of 40.5% (calcd. 42.45%) [103,104]. Finally, the loss of the remaining coordinated Hdmpz in the temperature range of 291–370 °C is observed with a weight loss of 24.4% (calcd. = 22.5%) [105].

### 3.5. Theoretical Studies

This section presents a theoretical investigation into the non-covalent interactions observed within the solid state of both compounds and their significant influence on X-ray structural properties. This analysis encompasses an examination of weak non-covalent interactions, including π-stacking and C-H···π interactions, as depicted in Figures 3 and 7. Furthermore, this study delves into hydrogenbonding interactions, which play a pivotal role in the formation of distinctive 1D supramolecular assemblies in both compounds, detailed in Figure 8. A comparative analysis of the energetic attributes of C-H···O, N-H···O, and O-H···O hydrogenbonding interactions is also conducted to enrich our understanding of their structural significance.

We calculated the molecular electrostatic potential (MEP) surfaces for compounds **1** and **2** to identify their most electrophilic and nucleophilic regions. The MEP surfaces, illustrated in Figure 9, reveal that for compound **1**, the MEP maximum is found at the Hatoms of the water molecule bound to the compound, with a value of 65.9 kcal/mol, closely followed by the -NH group (61.5 kcal/mol) of the Hdmpz ligand. These large and positive MEP values are attributed to the increased acidity of the OH$_2$ and -NH protons when coordinated to the Mn(II) metal center, suggesting that the water molecule acts as a more effective H-bond donor compared to the pyrazole ring. The MEP's lowest value occurs at the uncoordinated Oatom of the benzoate ligand. Positive MEP values are seen at

the Hatoms of methyl groups (approximately 18.8 kcal/mol), while negative values are associated with the benzoate and pyrazole $\pi$-systems ($-23.8$ kcal/mol and $-8.2$ kcal/mol, respectively). In contrast, for compound **2**, the highest MEP value is at the -NH group (51.5 kcal/mol) of the Hdmpz ligand, with the lowest at the Oatom of the crotonate ligand ($-53.3$ kcal/mol). The $\pi$-system of the double bond exhibits a significantly negative MEP, whereas the MEP at the Hatom of the methyl group is positive (18.9 kcal/mol) and slightly positive at the pyrazole ligand's $\pi$-system (6.3 kcal/mol), in sharp contrast to compound **1**.

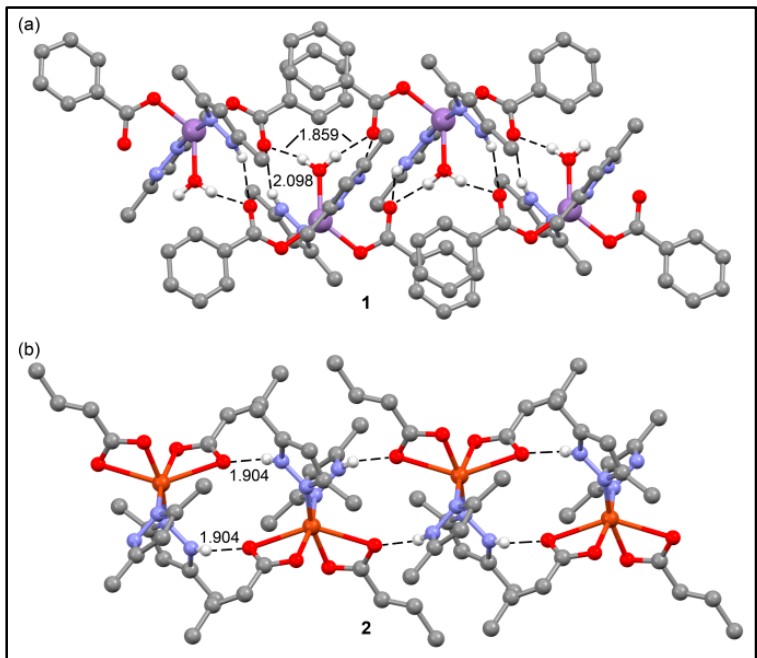

**Figure 8.** Partial view of the 1D infinite supramolecular chains observed in the solid state of compounds **1** (**a**) and **2** (**b**). Distances are in Å. The Hatoms have been omitted, apart from those participating in the O–H···O and N–H···O H-bonds.

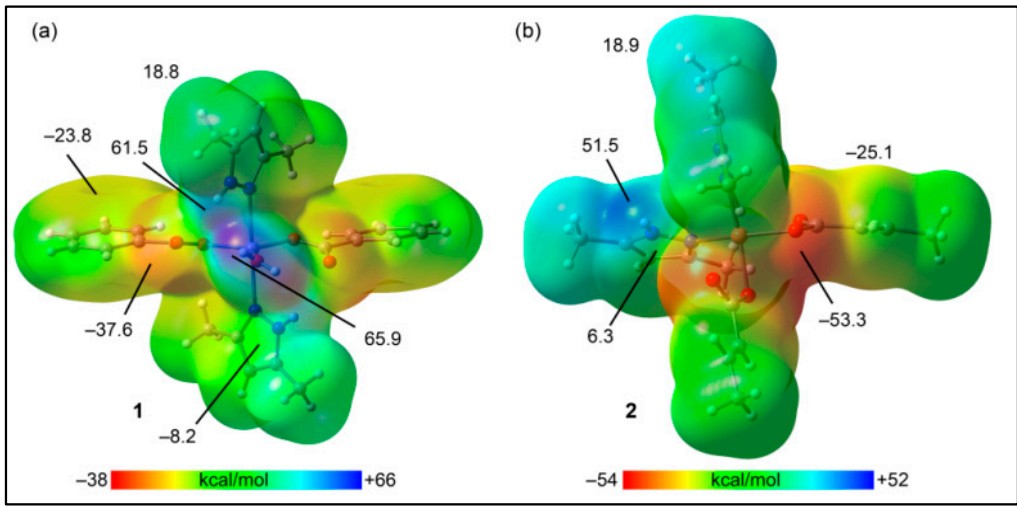

**Figure 9.** MEP surfaces of compounds **1** (**a**) and **2** (**b**). Isovalue: 0.001 a.u. Energies are given in kcal/mol.

In our study, we further investigated two dimeric configurations of compound **1**, as shown in Figure 10, to examine both hydrogen bonds (H-bonds) and $\pi$-interactions significant in the solid state, as referenced in Figures 3a and S1. Employing a combination of Quantum Theory of Atoms in Molecules (QTAIM) and Non-Covalent Interaction (NCI)

plot analyses allowed us to visually illustrate these interactions in real space. The NCI plot analysis uses color-coded reduceddensity gradient (RDG) isosurfaces to indicate the strength of interactions, with green and blue denoting weaker and stronger attractive forces, respectively.

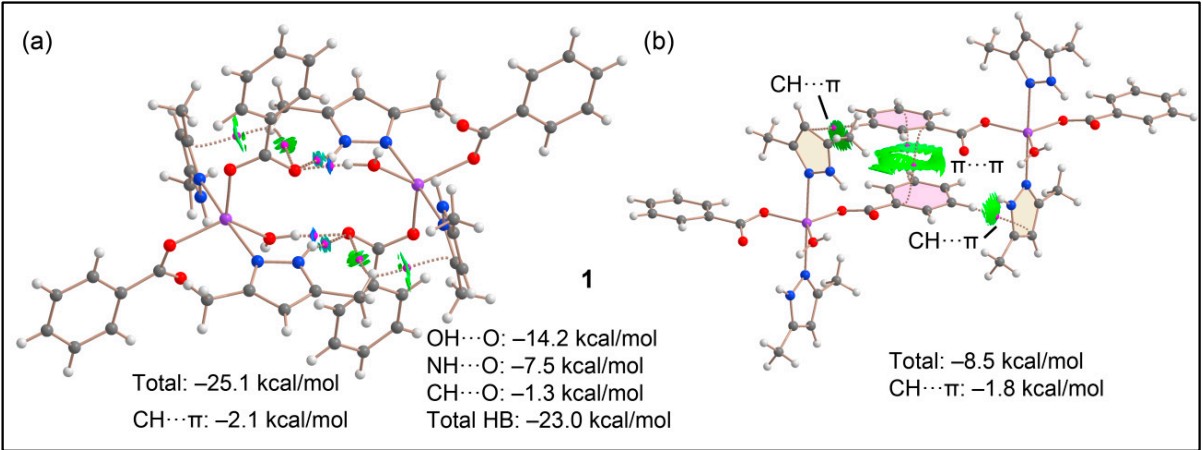

**Figure 10.** QTAIM (bond CPs in red and bond paths as dashed bonds) and QTAIM (RGD = 0.5, $\rho_{cut\text{-off}}$ = 0.035 a.u., color scale: $-0.035$ a.u. $\leq$ (sign$\lambda_2$)*$\rho \leq$ 0.035 a.u.) of H-bonded (**a**) and $\pi\cdots\pi$/CH$\cdots\pi$ (**b**) dimers of compound **1**. Only intermolecular interactions are represented.

Figure 10a presents the QTAIM/NCI plot analysis of a H-bonded dimer, extracted from the assembly depicted in Figures 3a and S1. It demonstrates that the uncoordinated Oatom of one monomer's benzoate ring forms three Hbonds with the adjacent monomer, stabilizing each dimer with a total of six H-bonds. These bonds are visualized through bond critical points (BCPs, marked as pink spheres), bond paths (depicted as dashed lines), and RDG disk-shaped isosurfaces. The RDG isosurfaces' color coding—dark blue for O–H$\cdots$O, light blue for N–H$\cdots$O, and green for C–H$\cdots$O interactions—indicates their respective strengths as strong, moderately strong, and weak. This aligns with the molecular electrostatic potential (MEP) surface analysis, confirming O–H$\cdots$O H-bonds as the strongest. The QTAIM data at the BCPs provide an estimation of each H-bond's contribution, with O–H$\cdots$O H-bonds contributing the most ($-14.2$ kcal/mol), followed by N–H$\cdots$O ($-7.5$ kcal/mol), and C–H$\cdots$O ($-1.3$ kcal/mol). The cumulative H-bond contribution is $-23.0$ kcal/mol, closely matching the total binding energy of $-25.1$ kcal/mol, with the discrepancy largely due to C–H$\cdots\pi$ interactions with the pyrazole, assessed at $-2.1$ kcal/mol by the QTAIM/NCI plot analysis.Conversely, Figure 10b illustrates a significant green RDG isosurface between the $\pi$-clouds of the coordinated benzoate rings, indicating $\pi$-stacking interactions. These interactions are supported by two BCPs and bond paths connecting the carbon atoms of both aromatic rings, suggesting a moderately strong binding energy ($-8.5$ kcal/mol). The QTAIM/NCI plot analysis also reveals two symmetrically equivalent C–H$\cdots\pi$ interactions, evidenced by their BCPs, bond paths, and green RDG isosurfaces, contributing $-1.8$ kcal/mol. This suggests the dominance of $\pi$-stacking interactions in this dimer configuration.

For compound **2**, our DFT analysis is centered on a hydrogen-bonded (H-bonded) dimer extracted from the one-dimensional (1D) infinite chain depicted in Figure 6a, highlighting its significance in the molecular packing of compound 2. The N-H$\cdots$O hydrogen bonds in this dimer are characterized through bond critical points (BCPs), bond paths, and dark blue reduced density gradient (RDG) isosurfaces. These features affirm the robustness of these interactions, as illustrated in Figure 11.

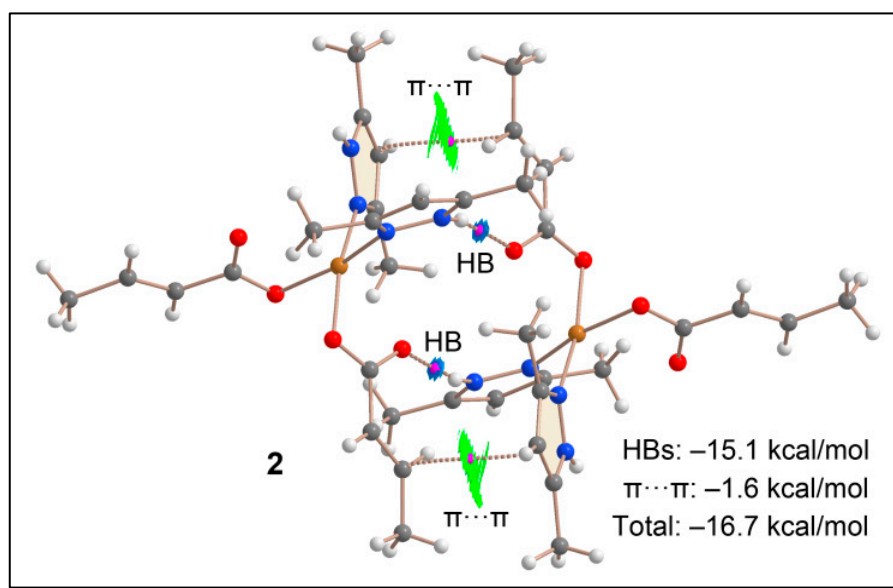

**Figure 11.** QTAIM (bond CPs in red and bond paths as dashed bond lines) and QTAIM (RGD = 0.5, $\rho_{\text{cut-off}}$ = 0.035 a.u., color scale: $-0.035$ a.u. $\leq$ (sign$\lambda_2$)*$\rho \leq$ 0.035 a.u.) of the assembly of compound 2. Only intermolecular interactions are represented.

The substantial strength of these interactions is further supported by the hydrogen bond (H-bond) energy, recorded at $-15.1$ kcal/mol. As illustrated in Figure 11, a notable RDG isosurface is visible between the $\pi$-acidic surface of the coordinated pyrazole ring and the $\pi$-basic double bond of the crotonate. This observation is in concordance with the MEP surface analysis. The interaction is additionally characterized by a BCP and a bond path that links one carbon atom of the double bond to a carbon atom of the aromatic ring. The overall binding energy of this dimer is evaluated as moderately strong, at $-16.7$ kcal/mol, predominantly influenced by the H-bonds, as the contribution from the $\pi\cdots\pi$ interaction is relatively minimal, amounting to $-1.6$ kcal/mol.

## 4. Conclusions

The synthesis of two new Mn(II) and Cu(II) coordination compounds, namely [Mn(bz)$_2$(Hdmpz)$_2$(H$_2$O)] (**1**) and [Cu(crot)$_2$(Hdmpz)$_2$] (**2**), was performed and characterized via a single crystal X-ray diffraction technique, FT-IR, electronic spectroscopy, TGA, and elemental analyses. Compound **1** is a penta-coordinated Mn(II) mononuclear compound, but compound **2** crystallizes as a hexa-coordinated Cu(II) compound of *Hdmpz*. The crystal structure analysis of compound **1** reveals the existence of C–H$\cdots\pi$ (2.70 Å, $-2.1$ kcal/mol) and $\pi$-stacking interactions (3.62 Å, $-6.7$ kcal/mol), which stabilizes the layered architecture of the compound along with the dominant O–H$\cdots$O (1.86 Å, $-14.2$ kcal/mol), N–H$\cdots$O (2.10 Å, $-7.5$kcal/mol), and C–H$\cdots$O (2.95 Å, $-1.3$ kcal/mol) H-bonding interactions. The presence of aromatic $\pi$-stacking ($-1.6$ kcal/mol), along with non-covalent C–H$\cdots$O (3.07 Å) and N–H$\cdots$O (1.90 Å) hydrogen bonding interactions ($-15.1$ kcal/mol) stabilizes the crystal structure of compound **2**. Theoretical study has delved into the non-covalent interactions in compounds **1** and **2**, focusing on hydrogen bonds in solid-state structures and $\pi$ interactions ($\pi\cdots\pi$ and C–H$\cdots\pi$). Molecular electrostatic potential (MEP) surfaces show that the coordinated water molecule and –NH group of *Hdmpz* are the primary H-bond donors. QTAIM and NCI plot analyses highlighted the nature and strength of these interactions. The results confirmed strong O–H$\cdots$O and N–H$\cdots$O hydrogen bonds and much weaker C–H$\cdots$O, $\pi\cdots\pi$ and C–H$\cdots\pi$ interactions. The synthesized compounds may find potential applications in the field of biology, as anti-bacterial agents, anticancer agents, etc. [106,107]. These compounds may also be potential candidates for homogenous catalysis [108].

**Supplementary Materials:** The following supporting information can be downloaded at https: //www.mdpi.com/article/10.3390/cryst14040318/s1, Figure S1: 1D chain of compound **1** involving intermolecular C–H···π and π-stacking interactions along the crystallographic c axis; Figure S2: 1D supramolecular chain of compound **2** assisted by non-covalent C–H···C interactions along the crystallographic b axis; Figure S3: FT-IR spectra of compounds **1** and **2**; Figure S4: (a) UV-Vis-NIR spectrum of **1**, (b) UV-Vis spectrum of **1**; Figure S5: (a) UV-Vis-NIR spectrum of **2**, (b) UV-Vis spectrum of **2**; Figure S6: Thermogravimetric curves of compounds **1** and **2**.

**Author Contributions:** Conceptualization, A.F. and M.K.B.; methodology, A.F. and M.K.B.; software, A.F.; formal analysis, A.F.; investigation, M.B. and T.B.; data curation, M.B.-O.; writing—original draft preparation, M.B., T.B. and M.K.B.; writing—review and editing, M.K.B.; visualization, M.K.B. and A.F.; supervision, M.K.B.; project administration, A.F. and M.K.B.; funding acquisition, A.F. and M.K.B. All authors have read and agreed to the published version of the manuscript.

**Funding:** Financial support was provided by SERB-SURE (Grant number: SUR/2022/001262), ASTEC, DST, Govt. of Assam (grant number ASTEC/S&T/192(177)/2020-2021/43) and the Gobierno de Espana, MICIU/AEI (project number PID2020-115637GB-I00), all of whom are gratefully acknowledged. The authors thank IIT-Guwahati for the TG data.

**Data Availability Statement:** Data are contained within the article and Supplementary Materials.

**Conflicts of Interest:** The authors declare no conflicts of interest. The funders had no role in the design of the study; in the collection, analyses, or interpretation of data; in the writing of the manuscript; or in the decision to publish the results.

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
