# Peer review of "Energetic Features of H-Bonded and π-Stacked Assemblies in Pyrazole-Based Coordination Compounds of Mn(II) and Cu(II): Experimental and Theoretical Studies"

_crystals, doi:10.3390/cryst14040318_

Round 1

Reviewer 1 Report

Comments and Suggestions for Authors

In my opinion the work is very good.

I just have a few minor comments:

1. Editorial error - line 134

2. Please standardize the way of marking the interactions shown in the drawings (lines or arrows)

3. Figure 2 is not very clear, perhaps for a smaller number of molecules the marked interactions would be more visible

4. In the conclusions, I would see some numerical data supporting the conclusions

Author Response

Thank you for your careful reading of the manuscript, corrections and suggestions. Our point-by-point responses follow:

In my opinion the work is very good.

I just have a few minor comments:

Q1. Editorial error - line 134

Reply: We have corrected the editorial error - line in the revised manuscript.

Q2. Please standardize the way of marking the interactions shown in the drawings (lines or arrows)

Reply: We have standardized the way of marking the interactions shown in the drawings by using lines in the revised manuscript.

Q3. Figure 2 is not very clear, perhaps for a smaller number of molecules the marked interactions would be more visible

Reply: We have tried to make the Figure 2 clear by showing the marked interactions for a smaller number of molecules in the revised manuscript.

Q4. In the conclusions, I would see some numerical data supporting the conclusions

Reply: We have incorporated some numerical data supporting the bonding distances and energetic significances of the interactions in the revised manuscript.

Reviewer 2 Report

Comments and Suggestions for Authors

The manuscript by Bhattacharyya and co-workers reports the syntheses and detailed characterization of pyrazole and carboxylate based coordination compounds. The authors have used a variety of characterization techniques to support their findings and have further evaluated the intramolecular interactions in detail. The manuscript may be accepted for publication in Crystals, however, I would appreciate if the authors address and take into account the following concerns;

1. It would be interesting to know that why did the authors use crotonate instead of benzoate to synthesize Mn complex? 

2. Comparing the bond distances of Cu complexes, the relative work of Khan and co-workers might be cited (Khan, E.; Gul, Z.; Shahzad, A.; Jan, M.S.; Ullah, F.; Tahir, M.N.; Noor, A. Coordination compounds of 4,5,6,7-tetrahydro-1H-indazole with Cu(II), Co(II) and Ag(I): Structural, antimicrobial, antioxidant and enzyme inhibition studies. J. Coord. Chem. 201770, 4054–4069).

3. Comparing the bond distances of Mn complexes, the relative work of  Singh, et al. might be cited. (Mononuclear manganese carboxylate complexes: synthesis and structural studies Polyhedron, 25 (18) (2006), pp. 3628-3638).

4. I wonder if the authors would be interested to study these complexes in terms of their potential applications. A short comment may be added to the manuscript.

5. Could the authors please comment on the low C value of compound 1 in elemental analysis? How about performing the PXRD analysis and comparing it to the single crystal X-ray analysis!

6. I believe that 53% of similarity index indicated by iThenticate is too high for a manuscript to be accepted for publication.

Comments on the Quality of English Language

Over all quality of the English is very good. However, there are certain minor mistakes that should be avoided in the revision. Such as but not limited to;

1. Line 16, "Cu(II) respectively containing.." should be, "Cu(II), respectively, containing....".

2. Line 17, " the presence C‒H⋯π..." should be, ' the presence of C‒H⋯π...'.

3. Line 32, "demands is still challenging" should be, "demands are still challenging".

4. Line 162, "and 1.82 BM respectively" should be, "and 1.82 BM, respectively,".

5. Some of the figure captions are missing full stop at the end of the text.

Author Response

Thank you for your careful reading of the manuscript, corrections and suggestions. Our point-by-point responses follow:

The manuscript by Bhattacharyya and co-workers reports the syntheses and detailed characterization of pyrazole and carboxylate based coordination compounds. The authors have used a variety of characterization techniques to support their findings and have further evaluated the intramolecular interactions in detail. The manuscript may be accepted for publication in Crystals, however, I would appreciate if the authors address and take into account the following concerns;

Q1. It would be interesting to know that why did the authors use crotonate instead of benzoate to synthesize Mn complex?

Reply: Here in this study, we used benzoate to synthesize the Mn(II) complex while we used crotonate to synthesize the Cu(II) complex. We also tried the vice versa but did not obtain single crystals of the desired complexes. The intension behind choosing such carboxylate anions was in their ability to bind with the metal centre through varying coordination modes and resulting in the formation of diverse structural topologies having potential use in biology. Although we have constraint our present study only to the crystal engineering of the compounds and characterization of the various non-covalent interactions through theoretical studies, we intend to focus on their biological aspects in the near future.

Q2. Comparing the bond distances of Cu complexes, the relative work of Khan and co-workers might be cited (Khan, E.; Gul, Z.; Shahzad, A.; Jan, M.S.; Ullah, F.; Tahir, M.N.; Noor, A. Coordination compounds of 4,5,6,7-tetrahydro-1H-indazole with Cu(II), Co(II) and Ag(I): Structural, antimicrobial, antioxidant and enzyme inhibition studies. J. Coord. Chem. 2017, 70, 4054–4069).

Reply: We have incorporated the stated reference in the revised manuscript.

Q3. Comparing the bond distances of Mn complexes, the relative work of  Singh, et al. might be cited. (Mononuclear manganese carboxylate complexes: synthesis and structural studies Polyhedron, 25 (18) (2006), pp. 3628-3638).

Reply: We have incorporated the stated reference in the revised manuscript.

Q4. I wonder if the authors would be interested to study these complexes in terms of their potential applications. A short comment may be added to the manuscript.

Reply: We would like to investigate the potency of these complexes as anticancer agents and in catalysis in the future. A comment on their potential applications has been added in the revised manuscript.

Q5. Could the authors please comment on the low C value of compound 1 in elemental analysis? How about performing the PXRD analysis and comparing it to the single crystal X-ray analysis!

Reply: We have carried out the elemental analysis once again and found out that C value in the compound 1 was 55.90% which now terminates the observance of low C value in compound 1. It would be very difficult for us to obtain single crystals of our synthesized compounds in the stipulated short span of time to perform the PXRD analysis for comparative purpose. Also, we depend on nearby institutes for PXRD. This is for kind consideration of the esteemed reviewer.

Q6. I believe that 53% of similarity index indicated by iThenticate is too high for a manuscript to be accepted for publication.

Reply: We have tried to reduce the percentage of similarity index in the revised manuscript as far as practicable.

Comments on the Quality of English Language

Over all quality of the English is very good. However, there are certain minor mistakes that should be avoided in the revision. Such as but not limited to;

Q1. Line 16, "Cu(II) respectively containing.." should be, "Cu(II), respectively, containing....".

Reply: We have corrected the stated mistake in the revised manuscript.

Q2. Line 17, " the presence C‒Hπ..." should be, ' the presence of C‒Hπ...'.

Reply: We have corrected the stated mistake in the revised manuscript.

Q3. Line 32, "demands is still challenging" should be, "demands are still challenging".

Reply: We have corrected the stated mistake in the revised manuscript.

Q4. Line 162, "and 1.82 BM respectively" should be, "and 1.82 BM, respectively,".

Reply: We have corrected the stated mistake in the revised manuscript.

Q5. Some of the figure captions are missing full stop at the end of the text.

Reply: We have corrected the stated mistake in the revised manuscript.

Reviewer 3 Report

Comments and Suggestions for Authors

The paper of Manjit K. Bhattacharyya and co-authors is about carboxylate coordination compounds of Mn(II) and Cu(II) with 3,5-dimethylpyrazole and benzoate and crotonoate anions respectively. Authors were succeeded in crystal growing amenable for single crystal X-Ray analysis, determined its structures. It was shown that 3,5-dimethylpyrazole coordinated to copper and manganese carboxylates in a 1:2 ratio. Compounds are mononuclear complexes. Authors also revealed and investigated non-covalent interactions in a crystals of obtained complexes.

The article is written carefully, I did not mention major mistakes.

The choice of research objects is not entirely clear: in the introduction, the authors spoke about the importance of coordination polymers, as well as a lot of information about non-covalent interactions. However, the choice of carboxylate anions is not at all clear: benzoic acid, despite the presence of a π-system, rarely forms π-π interactions. Typically, pentafluorobenzoic or pentachlorobenzoic acid is used for this. The choice of crotonic acid is even more incomprehensible. It is likely that the authors first prepared these complexes and only then wrote an introduction about the importance of noncovalent interactions. It seems to me that the introduction needs to be expanded to make the choice of carboxylate anions clear.

I also did not find any data in the text of the article about depositing ref codes into the Cambridge Structural Database - this must be done.

Line 51 “coonect”

Line 73 “analyses”

Table 1 “Emprical”

Lines 120-124: “Then, the resulting solution was kept undisturbed in a refrigerator (below 4°C) for crystallization. Blue block shaped single crystals were obtained by the slow evaporation of the mother liquor after several days.” - What was the temperature in the refrigerator? An aqueous solution at temperatures below 4 will evaporate extremely slowly and the phrase “several days” here is probably mistakeable. It is need to check at what temperature the crystals were grown and from what solvent. Also, based on the experimental data, it is not clear how the authors separated the target product from sodium chloride

The Cambridge Structural Database contains information on manganese benzoate with four molecules of 3,5-dimethylpyrazole (ref code 1063753). I advise the authors to compare the structural data they obtained with those previously deposited.

After answering the above questions, I definitely recommend accepting this work for Crystals.

Author Response

Thank you for your careful reading of the manuscript, corrections and suggestions. Our point-by-point responses follow:

The paper of Manjit K. Bhattacharyya and co-authors is about carboxylate coordination compounds of Mn(II) and Cu(II) with 3,5-dimethylpyrazole and benzoate and crotonoate anions respectively. Authors were succeeded in crystal growing amenable for single crystal X-Ray analysis, determined its structures. It was shown that 3,5-dimethylpyrazole coordinated to copper and manganese carboxylates in a 1:2 ratio. Compounds are mononuclear complexes. Authors also revealed and investigated non-covalent interactions in a crystals of obtained complexes.

The article is written carefully, I did not mention major mistakes.

Q1. The choice of research objects is not entirely clear: in the introduction, the authors spoke about the importance of coordination polymers, as well as a lot of information about non-covalent interactions. However, the choice of carboxylate anions is not at all clear: benzoic acid, despite the presence of a π-system, rarely forms π-π interactions. Typically, pentafluorobenzoic or pentachlorobenzoic acid is used for this. The choice of crotonic acid is even more incomprehensible. It is likely that the authors first prepared these complexes and only then wrote an introduction about the importance of noncovalent interactions. It seems to me that the introduction needs to be expanded to make the choice of carboxylate anions clear.

Reply: We have incorporated the reason behind using benzoate and crotonate anions for our present study. We have also expanded the introduction with respect to our choice of using carboxylate anions in terms of their diverse binding modes resulting intricating structural topologies and their potential biological applications in our revised manuscript.

Q2. I also did not find any data in the text of the article about depositing ref codes into the Cambridge Structural Database - this must be done.

Reply: We have provided the data regarding the deposition of ref codes into the Cambridge Structural Database in the revised manuscript.

Line 51 “coonect”

Reply: We have corrected this spelling error in the revised manuscript.

Line 73 “analyses”

Reply: We have corrected this spelling error in the revised manuscript.

Table 1 “Emprical”

Reply: We have corrected this spelling error in the revised manuscript.

Q3. Lines 120-124: “Then, the resulting solution was kept undisturbed in a refrigerator (below 4°C) for crystallization. Blue block shaped single crystals were obtained by the slow evaporation of the mother liquor after several days.” - What was the temperature in the refrigerator? An aqueous solution at temperatures below 4 will evaporate extremely slowly and the phrase “several days” here is probably mistakeable. It is need to check at what temperature the crystals were grown and from what solvent. Also, based on the experimental data, it is not clear how the authors separated the target product from sodium chloride

Reply: The temperature in the refrigerator was between 2-4 °C. The crystals were grown employing water as the solvent. Considering slow evaporation of the solvent at such low temperature we have obtained the crystals of our reported compounds after about one month. It is really difficult to be very specific on the temperature and the number of days required for getting single crystals. During crystallization, the pure crystals separated out of the solution leaving behind the other probable products in solution.

 Q4. The Cambridge Structural Database contains information on manganese benzoate with four molecules of 3,5-dimethylpyrazole (ref code 1063753). I advise the authors to compare the structural data they obtained with those previously deposited.

Reply: We have compared the structural data of our synthesized Mn(II) complex with previously deposited manganese benzoate complex with four molecules of 3,5-dimethylpyrazole (ref code 1063753) and found out that it has no structural similarity with our reported compound. We have now cited this report while comparing the average Mn-O and Mn-N bond distances in the revised manuscript.

 After answering the above questions, I definitely recommend accepting this work for Crystals.

Reviewer 4 Report

Comments and Suggestions for Authors

In this work,  the authors performed both experimental and computational studies for Mn(bz)2(Hdmpz)2(H2O)] and [Cu(crot)2(Hdmpz)2]. These transition metal complex consisting of Mn(II) and Cu(II) were synthesized and analyzed using various techniques including single crystal X-ray diffraction, FT-IR, electronic spectroscopy, TGA, and elemental analyses. Crystal structure analysis revealed the presence of the non-covalent interaction playing an important role in stabilizing its layered assembly. Theoretical investigations, including DFT calculations, were conducted to evaluate the energetics of different types of non-covalent interactions observed in the compounds. QTAIM and NCI plot index computational tools were utilized to characterize these interactions and assess the contribution of H-bonds.

This work can be interesting to both experimental and computational chemistry communities. I would like to ask the authors to consider the minor comments below.

1. page 4, line 146

“Using the Gaussian-16 program and the PBE0-D3 [60]/def2-TZVP level of theory”

Can the authors estimate the error of using incomplete basis set here? The def2-TZVP basis set is not optimal for transition metal calculations.

2. page 4, line 150

“non-covalent interaction plot (NCI Plot) [63] were used”

This work should also be cited as the reference for NCI plot: J. Chem. Theory Comput. 2011, 7, 3, 625–632.

3. page 11, Table 3

The meaning of numbers in the brackets is missing in the table.

4. page 12, Figure 8

The legend and unit are missing in the figure.

5. page 14, Figure 10

In the left subfigure, does it imply that pi-pi interactions give no contribution to  the stabilization?

Comments on the Quality of English Language

No major language or grammar problem found.

Author Response

Thank you for your careful reading of the manuscript, corrections and suggestions. Our point-by-point responses follow:

In this work, the authors performed both experimental and computational studies for Mn(bz)2(Hdmpz)2(H2O)] and [Cu(crot)2(Hdmpz)2]. These transition metal complex consisting of Mn(II) and Cu(II) were synthesized and analyzed using various techniques including single crystal X-ray diffraction, FT-IR, electronic spectroscopy, TGA, and elemental analyses. Crystal structure analysis revealed the presence of the non-covalent interaction playing an important role in stabilizing its layered assembly. Theoretical investigations, including DFT calculations, were conducted to evaluate the energetics of different types of non-covalent interactions observed in the compounds. QTAIM and NCI plot index computational tools were utilized to characterize these interactions and assess the contribution of H-bonds.

This work can be interesting to both experimental and computational chemistry communities. I would like to ask the authors to consider the minor comments below.

Q1. page 4, line 146. “Using the Gaussian-16 program and the PBE0-D3 [60]/def2-TZVP level of theory”. Can the authors estimate the error of using incomplete basis set here? The def2-TZVP basis set is not optimal for transition metal calculations.

Reply: For the first row of transition metals this basis set uses all electrons without ECP, so the error is expected to be minimal (below 1 kcal/mol). This basis set has been used previously by us and many other for transition metals providing reliable results.

Q2. page 4, line 150. “non-covalent interaction plot (NCI Plot) [63] were used”. This work should also be cited as the reference for NCI plot: J. Chem. Theory Comput. 2011, 7, 3, 625–632.

Reply: We have cited the stated reference for NCI plot in the revised manuscript.

Q3. page 11, Table 3, The meaning of numbers in the brackets is missing in the table.

Reply: We have removed the numbers in the brackets in the table to establish uniformity among all the data in the revised manuscript.

Q4. page 12, Figure 8. The legend and unit are missing in the figure.

Reply: Fixed, thanks

Q5. page 14, Figure 10. In the left subfigure, does it imply that pi-pi interactions give no contribution to the stabilization?

Reply: As in the left subfigure, no bond critical point along the bond path and green/blue isosurface is observed between the aromatic π systems, so it implies that pi-pi interactions give no contribution to the overall stabilization.

We thank esteemed reviewers for kind review and comments on our paper.